# Ridge Preservation and Augmentation Using a Carbonated Apatite Bone Graft Substitute: A Case Series

**DOI:** 10.3390/dj12030055

**Published:** 2024-02-28

**Authors:** Yoichi Taniguchi, Tatsuro Koyanagi, Toru Takagi, Yutaro Kitanaka, Akira Aoki, Takanori Iwata

**Affiliations:** 1Department of Periodontology, Graduate School of Medical and Dental Sciences, Tokyo Medical and Dental University (TMDU), Tokyo 113-8549, Japan; 2Oral Diagnosis and General Dentistry, Tokyo Medical and Dental University Hospital (TMDU), Tokyo 113-8549, Japan

**Keywords:** ridge preservation, ridge augmentation, carbonate apatite, implant, case report

## Abstract

The newly developed mineral carbonated apatite has recently been proposed as a bone graft material for bone regenerative treatment in implant therapy. This case series details the clinical and radiographic outcomes of ridge preservation and ridge augmentation using only carbonated apatite as bone graft material for implant treatment. Twenty patients (36 sites) who required bone regeneration and implant placement were retrospectively assessed. Simultaneous carbonated apatite implant placement was performed using the simultaneous ridge preservation or augmentation approach on 24 sites in 13 patients with sufficient bone quantity for primary stabilization based on preoperative evaluation results. A staged ridge preservation or augmentation approach was used for the remaining 12 sites in seven patients with insufficient bone quantity. The mean regenerated bone height for each treatment method was as follows: simultaneous preservation, 7.4 ± 3.3 mm; simultaneous augmentation, 3.6 ± 2.3 mm; staged preservation, 7.2 ± 4.5 mm; and staged augmentation, 6.1 ± 2.7 mm. The mean regenerated bone width for each treatment method was as follows: simultaneous preservation, 6.5 ± 2.9 mm; simultaneous augmentation, 3.3 ± 2.5 mm; staged preservation, 5.5 ± 1.7 mm; and staged augmentation, 3.5 ± 1.9 mm. Ultimately, the use of carbonated apatite alone as a bone graft material in implant therapy resulted in stable and favorable bone regeneration.

## 1. Introduction

Bone graft substitutes are used in periodontal regenerative therapy and peri-implant bone regeneration, such as ridge preservation (RP) and ridge augmentation (RA) therapy [1,2,3]. RP is a treatment to prevent the resorption of alveolar bone due to disuse atrophy after tooth extraction and to preserve the morphology of the remaining bone by placing bone grafting material in the extraction socket. RA is a treatment to increase the width and height of the resorbed bone due to post-extraction disuse atrophy to place implants in the correct position. Autogenous bone grafts are the “gold standard” for bone graft materials [4]. However, autogenous bone collection increases surgical invasiveness and operative time. In addition, the number of mesenchymal stem cells, which are effective in bone regeneration, is low in collected cortical bone, and fatty degeneration is often observed in cancellous bone among older adults [5]. For these reasons, the quality and quantity of the collected autogenous bone are unstable, and harvesting of the autogenous bone requires a donor site that is related to postoperative patient discomfort; thus, autogenous bone is not considered an ideal material for bone regeneration procedures in implant therapy [4]. Various bone graft substitutes, such as allograft and xenograft materials, have been applied in clinical practice [6,7,8]. The benefits of using bone graft substitutes include reduced surgical intervention and operative time and improved bone regeneration. In clinical practice, allografts, such as freeze-dried bone (FDBA) and demineralized FDBA (DFDBA), are commonly used; however, the effects of DFDBA and FDBA vary depending on the individual from which they were harvested. Bovine bone mineral (BBM) is a xenograft material currently used widely in bone regeneration therapy, with a substantial amount of clinical evidence supporting its use [6]. Moreover, because BBM is a standardized, non-resorbable biomaterial, it is hardly affected by individual differences; however, some reports have suggested that it does not induce complete remodeling of the bone [9,10].

Synthetic bone graft materials, such as hydroxyapatite (HA) and β-tricalcium phosphate (β-TCP), were adopted early for use in regenerative therapy; however, as HA is a non-resorbable material, bone remodeling hardly occurs, resulting in poor local stability [11]. In contrast, β-TCP provides better stability, although as an early resorbable material, the quantity of bone regeneration may not match the amount filled [12]. Carbonated apatite (CA) is a recently developed material with clinical application in bone regenerative therapy. CA has a composition similar to the major inorganic component of human bone. In contrast to synthetic bone graft materials, such as HA, CA is completely remodeled to the bone with a longer remodeling period than that of β-TCP. This delayed resorption characteristic is assumed to favor bone regeneration, as CA maintains the function as a scaffold for a longer period than β-TCP. Moreover, compared with BBM, early-stage new bone formation by CA granules has been observed in vivo, with a higher bone remodeling rate [13,14,15,16,17]. These qualities lead to better clinical outcomes following sinus floor elevation with implant placement compared to other synthetic bone graft materials [15]. CA is currently approved under the Japanese Pharmaceutical Affairs Act for use in periodontal regenerative therapy and peri-implant bone regeneration, such as RP and RA, and sinus floor elevation. Moreover, CA has been recognized as an osteopromotive bone substitute in some in vitro studies [14]. However, the use of CA materials for RP and RA therapies has not been reported. Therefore, this case series aimed to evaluate the clinical and radiographic outcomes of RP and RA when the novel material CA alone was used for implant treatment.

## 2. Materials and Methods

### 2.1. Case Series

The clinical data used in this case series were collected, de-identified, and analyzed. In this study, past medical records, intraoral photographs, and computed tomography (CT) image data of patients meeting the adoption criteria were extracted and numbered, and a linkable correspondence table was created. The data obtained were statistically analyzed. The cases included 36 implant sites of 20 patients (7 males and 13 females) requiring bone quantity adjustments, such as RP or RA, each with at least one edentulous area. This study was a retrospective study, and the inclusion criteria were based on a review of the past appointment registry and included 20 patients who had undergone implant treatment with RP and RA using CA alone from the year 2018 when CA was launched until 2022 and at least 2 years after prosthesis treatment. The exclusion criteria for surgery were psychiatric patients, smokers, pregnant women, and patients with systemic diseases that could affect the surgery (diabetes, cardiovascular disease, cerebrovascular disease, osteoporosis, taking bisphosphonate drugs, etc.). Exclusion criteria for this study included patients who could not be followed up for 2 years and patients who could not give consent for the study. The bone quantity of each implant site was insufficient for implant placement (Table 1). The CA product used in this case series is a material that has been approved by the Pharmaceutical Affairs Law in Japan, was used within the indication, and does not present any risks that need to be noted. All patients were informed about the risks associated with the procedures, and written informed consent was obtained from them. Clinical data were obtained from their medical records 12 months following bone regenerative therapy.

### 2.2. Case Series Design and Surgical Procedure

Bone regenerative treatment was planned using preoperative data obtained from cone-beam CT (CBCT) and wax-up study models. Each case was classified based on the bony housing morphology of the implant sites: intra-bony (for RP) or extra-bony (for horizontal and/or vertical RA) housing type. Based on the preoperative evaluation results in 36 sites of 20 patients, implant placement was simultaneously performed during RP (SiRP) or RA (SiRA) in 24 sites of 13 patients with sufficient bone quantity for primary stabilization. At the remaining 12 sites of seven patients with insufficient bone quantity, implant placement was performed using a staged approach, following RP (StRP) or RA (StRA) (Table 1). In all cases, bone regenerative therapy was conducted using only CA without guided bone membranes and regenerative agents, such as enamel matrix derivatives and other growth factors.

Since the whole procedure was estimated to require approximately 1 h, it was not necessary to use intravenous sedation and only submucosal and subperiosteal infiltration anesthesia was used, with 2–4 1.8 mL cartridges of xylocaine containing 1/80,000 epinephrine. The mucoperiosteal flap was reflected at the treatment sites under local anesthesia. For cases that required immediate implant placement, tooth extractions were performed after flap reflection and granulation tissues were completely removed using hand and/or rotary instruments. For cases that required simultaneous implant placement during RP or RA, implants were placed according to “top-down treatment planning”. After decortication and myelogenous blood collection, the CA (Cytrans Granules^®^, M size; GC Corporation, Tokyo, Japan) alone was mixed with the collected blood and grafted at the bone defect sites around each implant. When blood could not be collected, the CA was mixed with a saline solution. For RP, the CA was grafted to the whole length of the residual alveolar bone, whereas for RA, the CA was grafted beyond the residual bony housing. Wound closure was achieved following conventional, minimally invasive, periosteal-releasing incisions using a scalpel. All sutures were performed with interrupted sutures and 5–0 nylon was used. Postoperative medication was administered after confirming allergies, and only antibiotics (FLOMOX Tablets 100 mg; Shionogi, Osaka, Japan) and non-steroidal anti-inflammatory drugs (NSAIDs) were administered for 4 days. No allergies or other complications due to postoperative medication were observed. Postoperatively, both cases of RP and RA were prohibited from rinsing on the day of surgery. Brushing was discontinued until sutures were removed approximately 2 weeks later. In the case of RP, strong rinsing was prohibited for about 2 weeks due to the defect in the oral mucosa caused by teeth extraction. Suture removal was performed after adequate wound healing was confirmed at approximately 2–3 weeks postoperatively. Implant placement or uncovering was performed after 3–7 months. For StRP and StRA cases, each implant had sufficient primary stability (>20 N·cm). Uncovering was performed after 4–5 months. In every case, prosthetic delivery was performed without any complications, and maintenance was performed.

### 2.3. Clinical Measurements

In all cases, the dimensional changes in the regenerated peri-implant bone tissue were measured using baseline and postoperative clinical outcomes, such as CBCT, radiography, and intraoral photography (Figure 1). For the clinical measurement, two different methods were used to compensate for errors induced by CT artifacts. When using intraoral photographs, the most apical and coronal points of peri-implant bone shortage (apical point, a; coronal point, c) were determined using the images, and the distance between a and c was measured as regenerated bone height (RBH). The RBH of the simultaneous-approach cases was measured using intraoral photographs. For the CT images of the staged approach cases, implant placement simulations were performed using computer simulation software (Landmark system; iCAT, Osaka, Japan) on the pre-surgical CT image. Subsequently, post-surgical CT was performed to confirm the implant positions after 3–6 months following bone regenerative therapy. The simulated implant position on the pre-surgical CT image was reproduced on the post-surgical CT image, and the regenerated bone width (RBW) and RBH were measured. RBW was measured as the longest line perpendicular to the principal axis of the implant in the regenerated bone area, while RBH was defined as the longest line parallel to the principal axis of the implant. Changes in bone height and width were measured at each treatment site (Table 2). Buccolingual bone width was defined as the total thickness of the buccal and lingual/palatal bone width for RP or buccal bone width for RA without implant diameter.

Bone quantity achievement (BQA), regenerated bone quality (RBQ), and pre-/postoperative oral vestibular condition (OVC) were evaluated using a five-point scale during uncovering (Table 3 and Table 4). BQA was evaluated as the percentage of the predicted preoperative bone regeneration as follows: (1) very poor, less than 20%; (2) poor, approximately 40%; (3) moderate, approximately 60%; (4) good, approximately 80%; and (5) very good, approximately 100%. RBQ was assessed as follows: (1) very poor, no granules remodeled to the bone; (2) poor, granules were partially remodeled to the bone but a large number of granules remained; (3) moderate, granules were largely remodeled to the bone; (4) good, a few granules remained at the surface; and (5) very good, granules were completely remodeled to the bone. OVC was assessed as follows: (1) very narrow, free gingival graft treatment was needed during uncovering; (2) narrow, although free gingival graft treatment was not necessarily required, the flap required apical positioning; (3) moderate, soft tissue management was not necessary; (4) good, punch-out only; and (5) very good, required soft tissue reduction if necessary. In addition, post-surgical complications involving the peri-implant tissues were assessed.

Data were evaluated using standard statistical analysis software (R Foundation for Statistical Computing, Vienna, Austria). RBH and RBW were compared between RP and RA, Si and St, and blood-mixed CA and saline solution-mixed CA, using the following method. A linear mixed-effects model was fitted with fixed effects; an intercept with bone graft material as the fixed effect and patient as the intercept-variable effect was performed with the treatment procedure (Si and St), bone form (intra-bony housing type: RP and extra-bony housing type: RA), the interaction between the treatment procedure and bone form and mixed with bone graft material (blood and saline solution) as the fixed effect, and RBH and RBW as the ending variable. The minimum mean squares values were then calculated for each level of the fixed effect and inter-level comparisons (*p* < 0.01).

## 3. Results

In each case, all implants ensured osseointegration during uncovering, and the peri-implant hard tissue was deemed stable. The results for Cases 2 and 4 are shown in Figure 2 and Figure 3, respectively. For SiRP, after 5 months of RP treatment, the implant was placed in the regenerated bone up to the height of the preserved alveolar ridge (Figure 2). In Case 4, the bone was hard and only 7 mm remained from the alveolar ridge to the inferior alveolar canal. The minimum length of the implant in this case was 8 mm, and to avoid postoperative complications such as paralysis, a safety margin of 2 mm was established, and the implant was placed after drilling for 5 mm. In addition, the molar area had significant vertical bone resorption and narrowing of the oral vestibule, and it was predicted that brushing would be difficult after the prosthesis. Therefore, the plan was to perform a vertical alveolar crest augmentation. Approximately 4.0 mm of vertical bone regeneration was achieved at implant 46 at 4 months following RA. Additionally, the exposed implant threads appeared to be completely covered with regenerated bone in the panoramic radiograph taken 24 months after prosthetic delivery (Figure 3).

In most cases, the peri-implant bone was sufficiently regenerated, and the required bone height and width were recovered. The results of each procedure are shown in Table 2. For RP (n = 19), upon re-entry surgery after 4–7 months following RP, the mean RBH and regenerated buccolingual bone widths were 7.3 ± 3.7 and 6.2 ± 2.6 mm, respectively. For RA (n = 17), upon re-entry surgery after 4–7 months following RA, the mean RBH and regenerated buccolingual bone width were 4.5 ± 2.7 and 3.4 ± 2.3 mm, respectively. The RBW and RBH were significantly greater in the RP group than in the RA group. However, no significant differences were observed in the RBH and buccolingual bone width between the simultaneous and staged approaches or the blood-mixed CA and saline solution-mixed CA cases (Table 2).

The BQA, RBQ, and OVC medians and their total distributions are listed in Table 3. In most cases, the BQA and RBQ were rated above 4, and no changes were observed in the pre- and post-OVC. The BQA of SiRP and SiRA, StRP and StRA, and RBQ were rated 4, 3, and 4, respectively, in each procedure. The OVC was slightly different between the preoperative and postoperative stages (Table 4).

Postoperative complications, such as flap dehiscence, leakage of graft material, fistula, necrosis, purulence, and infection, were continually investigated after surgery (Table 5). No notable complications were observed during soft tissue wound healing after 3–7 months following surgery at all 36 sites. Although, postoperatively, mouth rinsing was prohibited, in seven RP cases, the patients performed strong rinsing, and the blood clots in the extraction socket fell out. After the loss of the blood clot, a small amount of exposed grafted bone (leakage of graft material) was observed after 1–2 weeks postoperatively. To address this problem, the exposed graft material was removed via irrigation with physiological saline, and extracted socket epithelization was confirmed 3–4 weeks postoperatively. The surgical site healed uneventfully.

## 4. Discussion

This case series demonstrated that CA is safe and effective in RP and RA procedures for implant placement. The preoperative examination in this study used the same criteria as Taniguchi et al. [18] in selecting the simultaneous or staged technique, and the surgical techniques (incision, implant placement, bone grafting, and suturing) were the same as in the literature. However, some complications (leakage of the grafted materials and soft tissue necrosis) were identified during follow-up. The absence of infection may be attributable to the non-porous structure of CA. Based on our clinical outcome, the regenerated bone quality and quantity were similar to those of cases without complications. However, it is possible that suturing of the extraction socket with a significant releasing incision might have been necessary to avoid leakage.

At the time of re-entry surgery, sufficient bone regeneration was determined using BQA and RBQ. Based on these results, a postoperative bone remodeling period of >5 months was recommended. In a report of sinus floor elevation using CA, human tissue samples harvested from regenerated bone were analyzed histologically, and sufficient ossification of CA was confirmed after 7 ± 2 months [17]. Although no human tissue samples were prepared in this study, the clinical results obtained suggest that the clinical outcomes were similar to previous reports. Compared to the rest of the literature, in the clinical studies using BBM, RA had implants placed after 6 months and RP had re-entry after 7 months [19,20]. The quality of regenerated bone was considered good in both studies. In the present study, CA showed clinical ossification at a relatively early stage of approximately 5 months. As a reason, it is thought that CA has an osteopromotive function [14]. Furthermore, in Japan, BBM is classified as a non-absorbable material, while CA is classified as a delayed resorption material. Therefore, the function of CA as a scaffold due to its delayed absorption was considered to be maintained, leading to good results. The result in only one patient was rated as very poor because of the cortical ossification of the residual bone wall caused by long-term inflammation due to root fracture. In this case, decortication was tried before CA grafting; however, there was no surrounding cancellous bone, and no bone marrow hemorrhage took place. Therefore, it is considered that the hemopoietic stem cells did not reach the CA sufficiently, resulting in delayed ossification.

The median BQA scores of the RA and RP groups were similar, whereas the RBW and RBH were significantly higher in RP cases than in RA cases. Notably, in RP, residual buccal bone protects the bone regeneration space from external stimuli. Conversely, in RA, the bone regeneration space with CA is compressed and narrowed by external stimuli; therefore, to induce new bone formation outside the bone house, the membrane technique or other procedures, such as block graft or the laser-assisted technique, are considered necessary [18,21,22]. Although some cases require the use of a membrane in RP, the gingival defect in the extraction socket may cause infection due to exposure to the membrane, and therefore, a membrane was not used in this study. In this study, RA cases showed sufficient bone width augmentation without a membrane; however, further major bone defects may require the use of a membrane in combination with RA. In this study, a control group was not prepared. Therefore, it was not possible to evaluate the difference in the materials used with the same technique. However, the clinical outcome of combining CA with these techniques has not been reported and should therefore be studied. At this point, it will be possible to compare the results with the clinical data obtained in this study when CA was used alone.

No significant differences in RBH and RBW or median BQA and RBQ values were observed between the blood- and saline solution-mixed CA groups. Based on actual values, the amount of bone regeneration was slightly higher in the blood-mixed group. However, it is difficult to accurately compare the amount of bone regeneration in volume using CT data, and at present, it is limited to measuring height and width. Bone quality is also measured using bone hardness and other data from surgical records, but more research is needed, including the collection of tissue sections. This was a retrospective study and patient data were not standardized. Therefore, preoperative bone morphology and requirements for bone regeneration are widely different, and postoperative statistics might be slightly unclear.

Regarding RP, van Steenberghe et al. reported in their study of SiRP that bone with a height of 7.0 mm was regenerated using BBM [20], which was comparable to our findings with CA. Regarding bone width, Barone et al. [19] showed that a mean ridge width of 8.1 ± 1.4 mm was preserved from the initial 10.6 ± 1.0 mm when BBM alone was used at the point of post-treatment evaluation, while less bone width was preserved using CA in our study. This disparity may be due to the differences in the original bone width among races. In the case of RA, Pelegrine et al. reported that the ridge width was augmented by 3.79 ± 0.52 mm using only BBM [23], similar to our findings with CA. It is difficult to directly compare results between individual case series owing to different clinical conditions; however, the above results suggest that CA has satisfactory effects comparable to that of BBM for RA and RP. Additionally, it is suggested that although CA and BBM have different basic structures—CA has a full structure, whereas BBM has a porous structure—they do not produce different clinical outcomes in terms of the amount of regenerated bone [24]. In this case series, retrospective data sampling was performed; therefore, the evaluation methods for bone regeneration and other parameters are limited. In the future, prospective clinical studies are required to obtain more accurate data. Moreover, the sample size of this clinical data is biased, and there is a necessity for more detailed and accurate data to be obtained from an even larger population, with a standardized sample size for each procedure. CA has only been available on the market for approximately 5 years, and only a few cases have confirmed long-term prognosis. However, there is a clinical realization from our clinical cases that regenerated bone is less likely to be lost than β-TCP [12] and that there are fewer granules remaining after about 4 months than in BBM. Therefore, good clinical outcomes are expected in large RA and RP treatments, and histological remodeling of CA to bone has been confirmed in sinus-floor elevation cases other than RP and RA [17,24]. Future follow-up and further study of these cases are necessary, as long-term and histological analyses of RP and RA are also needed.

In summary, our study showed that CA was successful in both RP and RA, that bone regeneration using CA is effective in the quantity and quality of bone regeneration, and that there were no postoperative incidental infections due to exposure to CA. This study is considered to have provided sufficient basic clinical data for the application of CA, which is expected to expand its field of use in the future.

## Figures and Tables

**Figure 1 dentistry-12-00055-f001:**
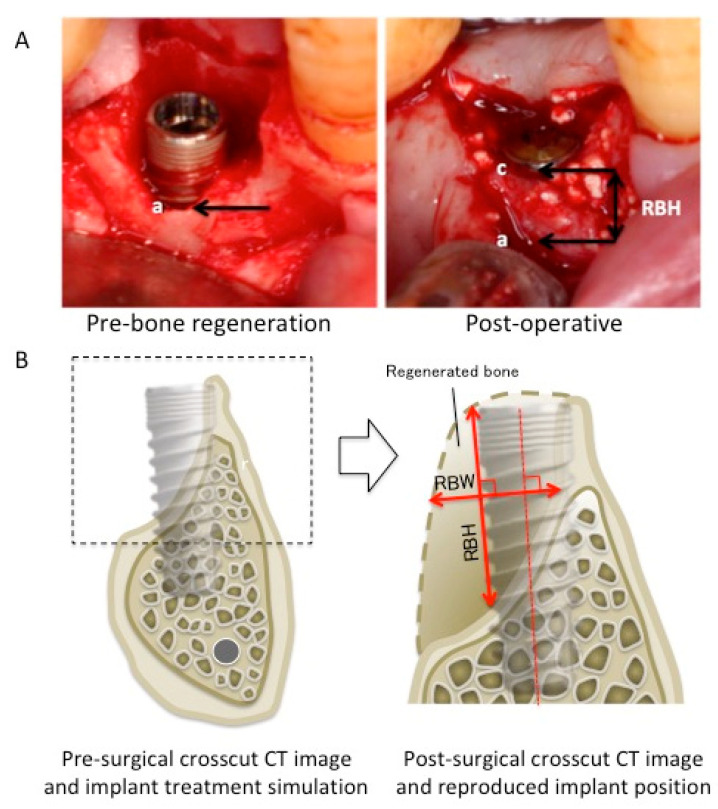
Clinical measurement. (**A**) An example of the oral photograph measurement: (a) original bone crest level and (c) postoperative bone height. (**B**) An example of the computed tomography measurement. RBW, regenerated bone width; RBH, regenerated bone height.

**Figure 2 dentistry-12-00055-f002:**
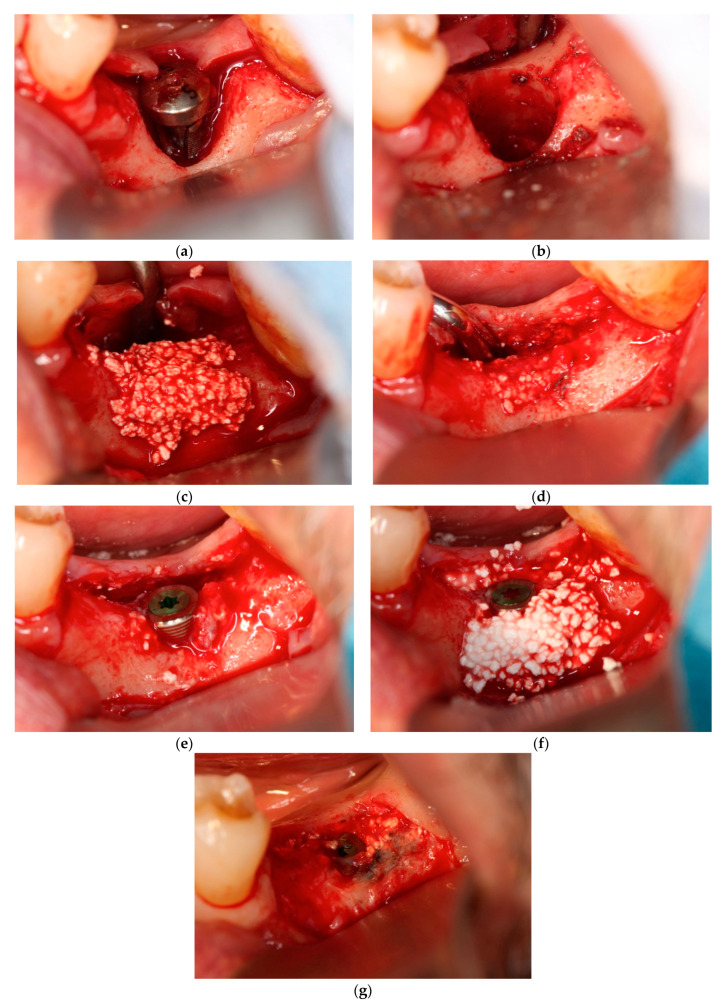
Case 2: Staged approach of implant placement following ridge preservation using carbonated apatite (CA; no membrane) immediately after fractured implant removal. (**a**) After flap reflection, a fractured and contaminated implant was observed. (**b**) Immediately after implant and granulation tissue removal. Residual bone volume was insufficient to achieve primary stabilization of simultaneous implant placement. (**c**) Bone graft. CA was placed into the implant removal socket. (**d**) After 5 months, at implant placement, the grafted CA was vertically and horizontally shrunk by external pressure. (**e**) Implant placement. The regenerated bone amount was not sufficient for the implant diameter. (**f**) Re-CA graft. CA was placed onto the exposed implant surface. (**g**) During uncovering. The new bone surrounding the implant was regenerated.

**Figure 3 dentistry-12-00055-f003:**
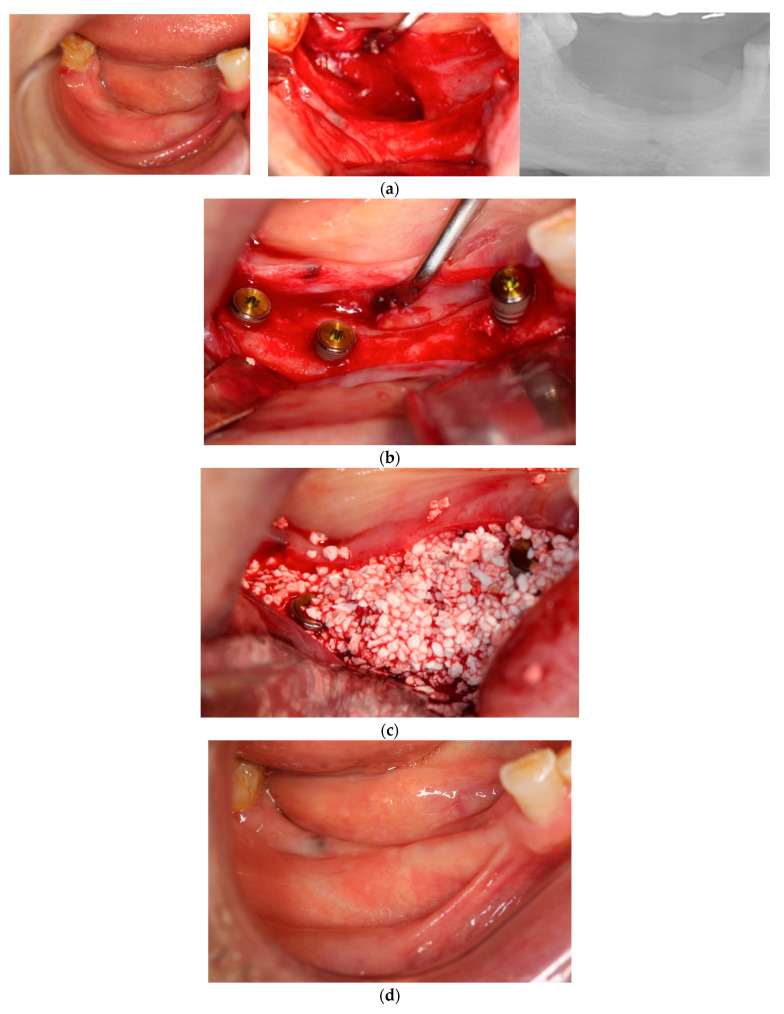
Case 4: Simultaneous approach of implant placement and vertical ridge augmentation using carbonated apatite (CA; no membrane). (**a**) Preoperative view. A vertical bone deficiency was observed at the mandibular right molar area on the panoramic radiograph. Soft tissue loss was also observed. (**b**) Implant placement. Implants were placed at the 44, 46, and 47 sites. Each implant achieved primary stability of over 30 N·cm. (**c**) CA graft. After a releasing incision to the lingual and buccal sites, CA was placed in the bone defect area to the platform level. (**d**) Immediately after treatment. Sutures were performed closely with interrupted sutures. (**e**) Five months after, before uncovering. Any complications were not observed. (**f**) Upon uncovering. Vertical and horizontal bone regeneration was observed at the mandible right molar area. The 44-implant thread was slightly exposed. (**g**) Twelve months after implant placement. Regenerated bone was maintained on a panoramic radiograph and no inflammation was observed in the peri-implant soft tissue.

**Table 1 dentistry-12-00055-t001:** Demographics of all patients and treated sites.

	Patients’ Demographics (N = 20)	Mean ± SD/N (%)
	Age (years)	57.9 ± 14.2
Sex	Male	7 (35%)
	Female	13 (65%)
	Treatment site (N = 36)	N(%)
Location of sites	Anterior mandible	6 (16.7%)
	Posterior mandible	12 (33.3%)
	Anterior maxilla	12 (33.3%)
	Posterior maxilla	6 (16.7%)
	Simultaneous approach	24 (66.7%)
	Ridge preservation	13 (36.1%)
	Ridge augmentation	11 (30.6%)
	Staged approach	12 (33.3%)
	Ridge preservation	6 (16.7%)
	Ridge augmentation	6 (16.7%)
Mixed with bone graft material	Blood	20 (55.6%)
	Saline solution	16 (44.4%)
Time to re-entry (months)		5.4 ± 1.1

SD, standard deviation.

**Table 2 dentistry-12-00055-t002:** Clinical parameters of bone regeneration upon re-entry surgery.

	Regenerated Bone Height (Mean ± SD mm)		Regenerated Bone Width (Mean ± SD mm)	
All procedure (N = 36)	6.0 ± 3.4			4.9 ± 2.7		
Simultaneous approach (N = 24)	5.7 ± 3.5	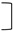	NS	5.0 ± 3.1	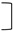	NS
Staged approach (N = 12)	6.6 ± 3.6	4.5 ± 2.0
Ridge preservation (N = 19)	7.3 ± 3.7	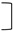	*	6.2 ± 2.6	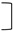	*
Ridge augmentation (N = 17)	4.5 ± 2.7	3.4 ± 2.3
Procedure						
SiRP (N = 13)	7.4 ± 3.4			6.5 ± 2.9		
SiRA (N = 11)	3.6 ± 2.3			3.3 ± 2.5		
StRP (N = 6)	7.2 ± 4.5			5.5 ± 1.7		
StRA (N = 6)	6.1 ± 2.7			3.5 ± 1.9		
Mixed with CA						
Blood (N = 20)	6.1 ± 2.7	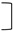	NS	4.7 ± 3.0	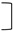	NS
Saline solution (N = 16)	5.8 ± 4.4	5.0 ± 2.5

SiRP, simultaneous approach of implant placement and ridge preservation; SiRA, simultaneous ridge augmentation; StRP, staged approach of implant placement following RP; StRA, staged ridge augmentation; SD, standard deviation; CA, carbonated apatite; NS, not significant; * *p* < 0.01, Welch’s *t*-test.

**Table 3 dentistry-12-00055-t003:** Evaluation of bone quality achievement, regenerated bone quantity, and oral vestibule condition.

	BQA (Median)	RBQ (Median)	Pre-OVC (Median)	Post-OVC (Median)
All procedure (N = 38)	4	4	3	3
Simultaneous approach	4	4	3	3
Staged approach	3	4	3	2.5
Ridge preservation	4	4	3	3
Ridge augmentation	4	4	3	3
Difference in Procedure				
SiRP	4	4	4	3
SiRA	4	4	3.5	3
StRP	3	4	3	2.5
StRA	3	4	3	2.5
Mixed with CA				
Blood	4	4	3	3
Saline solution	4	4	3	3

BQA of the predicted preoperative bone regeneration: (1) very poor, <20%; (2) poor, approximately 40%; (3) moderate, approximately 60%; (4) good, approximately 80%; and (5) very good, approximately 100%. RBQ: (1) very poor, granules remodeled to the bone were not observed at all; (2) poor, partially remodeled to the bone with a large number of granules remaining; (3) moderate, granules largely remodeled to the bone; (4) good, a few remaining granules at the surface; and (5) very good, granules completely remodeled to the bone. Pre- and Post-OVC: (1) very narrow, requires a free gingival graft during uncovering; (2) narrow, although a free gingival graft is not necessary, the flap requires apical positioning; (3) moderate, soft tissue management is unnecessary; (4) good, punch out only; and (5) very good, soft tissue reduction if necessary. CA, carbonated apatite; BQA, bone quantity achievement; RBQ, regenerated bone quality; Pre-OVC, preoperative oral vestibule condition; Post-OVC, postoperative oral vestibule condition.

**Table 4 dentistry-12-00055-t004:** Evaluation of bone quantity achievement, regenerated bone quality, and oral vestibule condition for the treatment procedures.

		BQR (Median)	RBQ (Median)	Pre-OVC (Median)	Post-OVC (Median)
All procedure		4	4	3	3
Simultaneous approach		4	4	3	3
Ridge preservation		3	4	4	2.5
Ridge augmentation		4	4	3	3
Difference in procedure		4	4	3	3
	SiRP	4	4	4	3
	SiRA	4	4	3.5	3
	StRP	3	4	3	2.5
	StRA	3	4	3	2.5
Mixed with CA					
	Blood	4	4	3	3
	Saline solution	4	4	3	3

SiRP, simultaneous ridge preservation; SiRA, simultaneous ridge augmentation; StRP, staged ridge preservation; StRA, staged ridge augmentation; SD, standard deviation; CA, carbonated apatite; BQA, bone quantity achievement; RBQ, regenerated bone quality; Pre-OVC, preoperative oral vestibule condition; Post-OVC, postoperative oral vestibule condition.

**Table 5 dentistry-12-00055-t005:** Postoperative complications.

Complications (N = 36)	N (%)
Flap dehiscence	0 (0%)
Leakage of graft material	7 (19.4%)
Necrosis	1 (2.8%)
Purulence	0 (0%)
Fistulas	0 (0%)
Infection	0 (0%)

N, number of treatment sites.

## Data Availability

The paper is self-contained. For additional information or data, please contact the corresponding author.

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
