# Peer review of "Ridge Preservation and Augmentation Using a Carbonated Apatite Bone Graft Substitute: A Case Series"

_dentistry, 2024, doi:10.3390/dj12030055_

Round 1

Reviewer 1 Report

Comments and Suggestions for Authors

Dear authors,

The manuscript presents a well-executed study on ridge preservation and augmentation utilizing carbonated apatite bone graft substitute demonstrated through a case series. The article is well-organized, and the language used is clear and accessible to a broad audience.

Overall, I believe that the manuscript is suitable for publication in the Dentistry Journal.

Good luck

Author Response

February 16th, 2024

Prof. Claude Jaquiéry

Editor-in-Chief

Dentistry Journal

Revision of manuscript “[Dentistry Journal] Manuscript ID: dentistry-2865456 ”

Dear Prof. Jaquiéry:

Thank you so much for your e-mail dated February 12, 2024 with regard to our manuscript (dentistry-2865456). We wish to express our appreciation to you and the reviewers for your valuable comments and suggestions. We are hereby sending you our revised manuscript based on those comments and suggestions,

In reply to your and the reviewers’ observations, we agree with your recommendations. We have considered your comments carefully and have made corrections and modifications (indicated by yellow highlights) to the manuscript which I hope will meet with your approval. Please read the revised manuscript and the following responses to the reviewers’ comments and suggestions.

Reviewer 1

Dear authors,

The manuscript presents a well-executed study on ridge preservation and augmentation utilizing carbonated apatite bone graft substitute demonstrated through a case series. The article is well-organized, and the language used is clear and accessible to a broad audience.

Overall, I believe that the manuscript is suitable for publication in the Dentistry Journal.

Good luck

Thank you for your review of our paper.

Reviewer 2 Report

Comments and Suggestions for Authors

Good Luck in future research!

Author Response

February 16th, 2024

Prof. Claude Jaquiéry

Editor-in-Chief

Dentistry Journal

Revision of manuscript “[Dentistry Journal] Manuscript ID: dentistry-2865456 ”

Dear Prof. Jaquiéry:

Thank you so much for your e-mail dated February 12, 2024 with regard to our manuscript (dentistry-2865456). We wish to express our appreciation to you and the reviewers for your valuable comments and suggestions. We are hereby sending you our revised manuscript based on those comments and suggestions,

In reply to your and the reviewers’ observations, we agree with your recommendations. We have considered your comments carefully and have made corrections and modifications (indicated by yellow highlights) to the manuscript which I hope will meet with your approval. Please read the revised manuscript and the following responses to the reviewers’ comments and suggestions.

Reviewer 2

Response and Guidelines for Authors on Review Article:

Ridge preservation and augmentation using carbonated apatite 2 bone graft substitute: a case series
A group of authors and specialists in the field of Tokyo Medical and Dental University Hospital (TMDU) from Japan brought to the light of research, the alternative of using CO3Ap as a bone graft substitute in ridge preservation and augmentation, all thise in exemples of clinical cases.

Dear Authors,
Congratulation for the Manuscript!
In what follows, I will specify some of the small adnotations:

Figure 2 and Figure 3:

Please reshaped in one figure the pictre a, b,....... with addnotation a,b, .....in the corner of the write picture used in Figure puzle (like in Figure 1)

We have improved the figures as noted.

Please build the Conclusion Chapter in front of Results,Statistics and Discution that are trully scientific and very good do it.

Thank you for your suggestion. My apologies - I am not certain of what you are asking. Are you suggesting we make a dedicated conclusion section at the end of the manuscript?

What is the Authors opinion about the using of Ca3Ap at large scale in bone regeneration? What is the prognostic?

The following text has been added to the discussion. It is highlighted in yellow. (line: 361-368)

“CA has only been available on the market for approximately 5 years, and only a few cases have confirmed long-term prognosis. However, there is clinical realization from our clinical cases that regenerated bone is less likely to be lost than β-TCP, and that there are fewer granules remaining after about 4 months than in BBM. Therefore, good clinical outcomes are expected in large RA and RP treatments, and histological remodeling of CA to bone has been confirmed in sinus-floor elevation cases other than RP and RA. Future follow-up and further study of these cases is necessary, as long-term and histological analyses of RP and RA are also needed.”

Have a good speed in reshapeing the Manuscript.

Thank you for your suggestion.

Reviewer 3 Report

Comments and Suggestions for Authors

I would like to suggest to the Authors to remove case report at the line 77...and replace with series.

In addiction I would like to recommend to the Authors to give a more detailed explanation to RP and RA correlated to the use of reabsorbable membranes.

Author Response

February 16th, 2024

Prof. Claude Jaquiéry

Editor-in-Chief

Dentistry Journal

Revision of manuscript “[Dentistry Journal] Manuscript ID: dentistry-2865456 ”

Dear Prof. Jaquiéry:

Thank you so much for your e-mail dated February 12, 2024 with regard to our manuscript (dentistry-2865456). We wish to express our appreciation to you and the reviewers for your valuable comments and suggestions. We are hereby sending you our revised manuscript based on those comments and suggestions,

In reply to your and the reviewers’ observations, we agree with your recommendations. We have considered your comments carefully and have made corrections and modifications (indicated by yellow highlights) to the manuscript which I hope will meet with your approval. Please read the revised manuscript and the following responses to the reviewers’ comments and suggestions.

Reviewer 3

I would like to suggest to the Authors to remove case report at the line 77...and replace with series.

We have improved this point. (line: 78)

In addiction I would like to recommend to the Authors to give a more detailed explanation to RP and RA correlated to the use of reabsorbable membranes.

We have added the following sentence. It is highlighted in yellow. (line: 323-327)

“Although some cases require the use of a membrane in RP, the gingival defect in the extraction socket may cause infection due to exposure of the membrane, and therefore a membrane was not used in this study. In this study, RA cases showed sufficient bone width augmentation without a membrane; however, further major bone defects may require the use of a membrane in combination with RA.”

Reviewer 4 Report

Comments and Suggestions for Authors

This scoping review presents the positive effects of orthodontic tooth movement on the periodontium in the grafted areas. It is well documented to be accepted in its present form. 

Author Response

February 16th, 2024

Prof. Claude Jaquiéry

Editor-in-Chief

Dentistry Journal

Revision of manuscript “[Dentistry Journal] Manuscript ID: dentistry-2865456 ”

Dear Prof. Jaquiéry:

Thank you so much for your e-mail dated February 12, 2024 with regard to our manuscript (dentistry-2865456). We wish to express our appreciation to you and the reviewers for your valuable comments and suggestions. We are hereby sending you our revised manuscript based on those comments and suggestions,

In reply to your and the reviewers’ observations, we agree with your recommendations. We have considered your comments carefully and have made corrections and modifications (indicated by yellow highlights) to the manuscript which I hope will meet with your approval. Please read the revised manuscript and the following responses to the reviewers’ comments and suggestions.

Reviewer 4

This scoping review presents the positive effects of orthodontic tooth movement on the periodontium in the grafted areas. It is well documented to be accepted in its present form. 

Thank you for your review of our paper.

Reviewer 5 Report

Comments and Suggestions for Authors

This manuscript is interesting because it introduces new graft material. However, in order for the paper to be published, there are several parts that need to be modified or added. This will provide good information to clinicians. I would like to give a major revision opinion on this manuscript.

1. #89 line: Replace “BP drug” with “bisphosphonate drug"

2. #110 line: Record the reason for not using a barrier membrane and compare it with other papers in the discussion. Bone graft is important, but the barrier membrane is also important.

3. #114 line: "1/8000 epinephrine" ---> Please check again.

4. #115 line: "Mucoperiosteal flap elevation was performed "---> Mucoperiosteal flap was reflected.

5.#116 line: "dental"--delete.

6. #117 line: "by" ---> using

7. #126 / #251 line: "simple" ---> interrupted

8. #127 line: "thread" ---> delete

9. #128 line: "antimicrobials" --->antibiotics (what type, what manufacturer)

10. #129-130 line: "The antimicrobial agents -----additionally prescribed" ------> Delete.

11. # 139 / #252 line: "second-stage operation" --->Replace with "uncovering". Please replace everything in the manuscript.

12. #140 / #231 line: prosthetic treatment ----> prosthetic delivery

13. #218 line: "osteointegration" ----->"osseointegration"

14.  #236 line: "infected"----> "contaminated"

15. #240 line: "quantity" ----> " amount"

16. #246: "shortage" -----> "deficiency"

17. #247: "shrinkage" ----> "loss"

18. #250: "bone shortage" ----> "bone defect"

19. #254: "insufficient" ----> "exposed"

20. #272 : "The incidence of" -----> "The postoperative"

21. #280 line : Replace with “The surgical site was healed uneventfully.”

22. #319 line : Replace “forces” with “stimuli”.

23. #336-#346 : Do not specifically mention the measured values in “Result” in “Discussion”. Please describe this part more simply.

24. #354  line: Describe the shortcomings of your paper. (1) Disadvantage of not having a control group, (2) Disadvantage of not having histological analysis, (3) Disadvantage of lack of long-term data...

Comments on the Quality of English Language

It is necessary to carefully look at the terminology (related to implant) used in the paper.

Author Response

February 16th, 2024

Prof. Claude Jaquiéry

Editor-in-Chief

Dentistry Journal

Revision of manuscript “[Dentistry Journal] Manuscript ID: dentistry-2865456 ”

Dear Prof. Jaquiéry:

Thank you so much for your e-mail dated February 12, 2024 with regard to our manuscript (dentistry-2865456). We wish to express our appreciation to you and the reviewers for your valuable comments and suggestions. We are hereby sending you our revised manuscript based on those comments and suggestions,

In reply to your and the reviewers’ observations, we agree with your recommendations. We have considered your comments carefully and have made corrections and modifications (indicated by yellow highlights) to the manuscript which I hope will meet with your approval. Please read the revised manuscript and the following responses to the reviewers’ comments and suggestions.

Reviewer 5

This manuscript is interesting because it introduces new graft material. However, in order for the paper to be published, there are several parts that need to be modified or added. This will provide good information to clinicians. I would like to give a major revision opinion on this manuscript.

Thank you for your review of our paper.

  1. #89 line: Replace “BP drug” with “bisphosphonate drug"

We have improved this point. The change is shown in yellow. (line: 90)

  1. #110 line: Record the reason for not using a barrier membrane and compare it with other papers in the discussion. Bone graft is important, but the barrier membrane is also important.

We have improved this point. The change is shown in yellow. (line: 323-327)

“ Although some cases require the use of a membrane in RP, the gingival defect in the extraction socket may cause infection due to exposure of the membrane, and therefore a membrane was not used in this study. In this study, RA cases showed sufficient bone width augmentation without a membrane; however, further major bone width defects may require the use of a membrane in combination with RA. ”

  1. #114 line: "1/8000 epinephrine" ---> Please check again.

We have improved this point. The change is shown in yellow. (line: 116)

  1. #115 line: "Mucoperiosteal flap elevation was performed "---> Mucoperiosteal flap was reflected.

We have made this change in yellow. (line: 117)

5.#116 line: "dental"--delete.

We have made this change.

  1. #117 line: "by" ---> using

We have made this change in yellow. (line: 119)

  1. #126 / #251 line: "simple" ---> interrupted

We have made this change in yellow. (line: 128, 254)

  1. #127 line: "thread" ---> delete

We have made this change.

  1. #128 line: "antimicrobials" --->antibiotics (what type, what manufacturer)

We have added the following in yellow. (line: 299, 343-350)

“…antibiotics (FLOMOX Tablets 100mg; Shionogi, Osaka, Japan)…”

  1. #129-130 line: "The antimicrobial agents -----additionally prescribed" ------> Delete.

We have made this change.

  1. # 139 / #252 line: "second-stage operation" --->Replace with "uncovering". Please replace everything in the manuscript.

We have made this change in yellow. (line: 137, 139, 174, 182, 195, 218, 243, 255)

  1. #140 / #231 line: prosthetic treatment ----> prosthetic delivery

We have made this change in yellow. (line: 140, 231)

  1. #218 line: "osteointegration" ----->"osseointegration"

We have made this change in yellow. (line: 218)

  1. #236 line: "infected"----> "contaminated"

We have made this change in yellow. (line: 237)

  1. #240 line: "quantity" ----> " amount"

We have made this change in yellow. (line: 241)

  1. #246: "shortage" -----> "deficiency"

We have made this change in yellow. (line: 249)

  1. #247: "shrinkage" ----> "loss"

We have made this change in yellow. (line: 250)

  1. #250: "bone shortage" ----> "bone defect"

We have made this change in yellow. (line: 253)

  1. #254: "insufficient" ----> "exposed"

We have made this change in yellow. (line: 257)

  1. #272 : "The incidence of" -----> "The postoperative"

We have made this change in yellow. (line: 274)

  1. #280 line : Replace with “The surgical site was healed uneventfully.”

We have made this change in yellow. (line: 283)

  1. #319 line : Replace “forces” with “stimuli”.

We have made this change in yellow. (line: 319,320)

  1. #336-#346 : Do not specifically mention the measured values in “Result” in “Discussion”. Please describe this part more simply.

I have simplified this section as per your request. (line: 299, 343-350)

  1. #354  line: Describe the shortcomings of your paper. (1) Disadvantage of not having a control group, (2) Disadvantage of not having histological analysis, (3) Disadvantage of lack of long-term data...

We have made this change in yellow. (line: 327-329, 361-368)

“In this study, a control group was not prepared. Therefore, it was not possible to evaluate the difference in the materials used with the same technique.”

“CA has only been available on the market for approximately 5 years, and only a few cases have confirmed long-term prognosis. However, there is clinical realization from our clinical cases that regenerated bone is less likely to be lost than β-TCP, and that there are fewer granules remaining after about 4 months than in BBM. Therefore, good clinical outcomes are expected in large RA and RP treatments, and histological remodeling of CA to bone has been confirmed in sinus-floor elevation cases other than RP and RA. Future follow-up and further study of these cases is necessary, as long-term and histological analyses of RP and RA are also needed.”

Round 2

Reviewer 5 Report

Comments and Suggestions for Authors

This manuscript has been well revised. It is suitable for publication.